# Effects of Socio-Familial Behavior on Sleep Quality Predictive Risk Factors in Individuals under Social Isolation

**DOI:** 10.3390/ijerph19063702

**Published:** 2022-03-20

**Authors:** Gilberto M. Galindo-Aldana, Luis A. Padilla-López, Cynthia Torres-González, Ibza A. García-León, Joaquín A. Padilla-Bautista, Daniel N. Alvarez-Núñez

**Affiliations:** 1Laboratory of Neurosciences and Cognition, Guadalupe Victoria Engineering and Business Faculty, Autonomous University of Baja California, Mexicali 21100, Mexico; gilberto.galindo.aldana@uabc.edu.mx (G.M.G.-A.); cynthia.torres.gonzalez@uabc.edu.mx (C.T.-G.); 2Laboratory of Psychophysiology, Human Sciences Faculty, Autonomous University of Baja California, Mexicali 21100, Mexico; alfredo.padilla@uabc.edu.mx; 3Psychology Master Program, CETyS University, Mexicali 21259, Mexico; ibza.garcia@cetys.mx; 4Laboratory of Psychosocial Research, Guadalupe Victoria Engineering and Business Faculty, Autonomous University of Baja California, Mexicali 21100, Mexico; joaquin.padilla@uabc.edu.mx; 5Neuropsychology Master Program, CETyS University, Mexicali 21259, Mexico

**Keywords:** pandemic, social isolation, families, sleep health, COVID-19

## Abstract

Social confinement involves a series of temporary changes in the habits and lifestyles of individuals, severely affecting their regular activities and schedules and substantially modifying socio-familial behavior (SFB) and sleep quality (SQ). There is no literature reporting the effects of SFB changes on SQ during social confinement due to the COVID-19 outbreak. An observational transversal research design, with group comparison and correlation methods, was used to perform the present study. The results were analyzed as follows: (1) An exploratory factor analysis (EFA); (2) A description of the sample was determined by proportions comparisons of sleep habits between the different variables of interest; and (3) A linear regression model was analyzed to explore the predictive association of the negative effects of social isolation during the COVID-19 pandemic on SFB and SQ. In addition to the global SFB score, two SFB factors were identified as predictors affecting the SQ, SF-Habits, and SF-Emotional scores, suggesting a close balance between daily life activities and sleep health during critical social changes. Furthermore, two main risk factors resulted from the regression analysis: economic concerns and increased alcohol consumption. Therefore, the predictive capacity of economic concerns showed statistical significance in anticipating negative sleep quality scores. Overall, this suggests that sleep quality, economic concerns, schedules, and substance use were associated with the self-perception of coping skills, elucidating the importance of fostering habits related to schedules within the home and ensuring that all family members participate.

## 1. Introduction

Social confinement involves a series of temporary changes in habits and lifestyles among families [1], which directly affects their regular activities, schedules, interactions between parents and children, sleep, and social behaviors [2] as mental health-related factors [3]. In December 2019, the World Health Organization declared the coronavirus disease (COVID-19) a public health emergency of international concern, which made it necessary to implement sanitary measures worldwide to reduce the number of cases and the risk of contagion. Among these measures, voluntary social confinement has been one of the most important and has substantially changed human lives. In some countries, the majority of the population started to isolate themselves at home [4], while others continued to have partial to no social isolation. Confirmed positive SARS-CoV-2 patients represent an important risk for health and the dissemination of the disease, as well as a challenge for health services and the workforce. However, there is a large population that experiences potential secondary effects on health and mental health as possible consequences of social confinement. During the acute phase of confinement, a large percentage of the populations of different countries around the globe maintained their daily routines since their jobs consisted of primary and essential services, but the rest of the individuals remained inside their homes after the outbreak. Most of the recent studies conducted in isolated populations of different countries and previous studies of groups of people in similar conditions reported negative psychological and physiological effects, mainly including depression, anxiety, and stress symptoms [5,6,7,8]. There is no literature reporting risk factors for SQ due to changes in SFB during social confinement due to the COVID-19 outbreak.

On one hand, family members’ interactions depend on the characteristics of the structure of the family, including whether there are children at home [9,10], and on the other hand, on the type of interactions, including emotional [10] or functional [11,12] interactions, and represents the main priority for surviving in several social organizations, which may be closely related to other life health factors.

Differences in demographics and socioeconomic status lead to variations in behavioral factors in groups facing pandemic outbreaks, specifically affecting psychological and socio-familial factors [1]. Self-isolation due to an outbreak combined with other related factors, such as economic burden, family and social support, fear of becoming infected, and the constant uncertainty about the current situation, for such a long period of time has never been seen before, which is why there is limited information about the impact of the aforementioned factors on sleep patterns [13,14].

Confinement represents a risk for different dimensions of the behavior of humans and can be observed in personal and familial effects, primarily sleep habits, and secondarily socio-familial behaviors. In addition, during health-related disasters, social effects may affect social, familial, and economic subgroups differently [15]. Arber and collaborators [16] studied the association between sociodemographic and socioeconomic factors and sleep problems in a British population. Under normal conditions, they found that 10% of the individuals in the highest household income group reported sleep complaints to some degree, compared to 25% of those in the lowest group; 12% of adults with some degree of education reported sleep problems, compared to 22% of people with no qualifications; for those who were employed full-time, 12% reported complaints, compared to 25% of those who were unemployed or economically inactive; and 15% of homeowners reported complaints, compared to 26% of those who paid some kind of rent.

Other research has described some possible negative sleep-related consequences of social confinement due to COVID-19 in a European population [17]. In their findings, the authors reported that the main risk factors for sleep problems were financial worries for entrepreneurs and small business owners, the inability to engage in rewarding activities, spending more time with family in small spaces, pre-existing conflicts among family members, the exacerbation of loneliness when living alone and the increase in childcare and household responsibilities.

Simultaneously, sleep habits can be altered by the consumption of certain substances, such as caffeine and alcohol. Day schedules, work habits, or home routines may become mismatched due to social isolation, may lead to an increase in alcohol consumption [18], and can be associated with circadian misalignment and sleep loss [19]. It is estimated that more than one in ten individuals consume alcohol as a hypnotic agent to self-medicate for sleep problems [20]. Since alcohol works as a central nervous system inhibitor, the effects of an acute pre-bedtime dose on sleep have been widely studied in different populations according to sex, age, the timing of intake, etc. A commonly reported phenomenon is shortened sleep onset, but when alcohol concentrations in the blood start to fall, sleep is disrupted with increased wakefulness [21,22,23,24]. When the consumption of alcohol becomes frequent, this pattern of initial sleep augmentation followed by poor sleep quality can lead to insomnia [25]. Another way in which the sleep–wake cycle (SWC) can be disrupted is through the presence of behaviors related to poor sleep hygiene, including irregular bedtimes, frequent or prolonged daytime naps, and staying in bed for non-sleep activities [26]. These alterations occur because the system that regulates the circadian rhythm has complex interactions with daily behaviors, and these interactions are known as entraining factors [27]. It has been reported that having a regular bedtime schedule can strengthen the circadian rhythm, and it is beneficial for achieving good quality sleep. It is suggested that a single night of sleep modifications may be sufficient to induce difficulties in initiating and maintaining sleep [28]. Prather and collaborators [29] studied a sample of 18,779 adults in the US, with the purpose of relating sleep patterns and carbonated and noncarbonated sugar-sweetened beverage (SSB) consumption, finding that individuals who slept 5 h or less consumed a significantly greater amount of SSBs than those who slept 7 or 8 h per night. Experimental studies have proposed the increase in intake of these types of beverages as a consequence and not necessarily a cause of sleep disturbances, since it was observed that adults exposed to a restriction of sleep hours displayed alterations in satiety hormonal regulators that cause hunger at nonconventional hours and preferences for high-fat, carbohydrate-rich foods [30,31,32].

Due to the lack of reliable data for sleep quality measures for populations under social self-isolation during an outbreak, the first stage of this study was resolved to develop and validate a specific questionnaire for measuring sleep quality. Second, we aimed to describe the possible effects of socio-familial behavior on sleep quality in a sample of participants under social self-isolation by the COVID-19 outbreak.

## 2. Materials and Methods

### 2.1. Study Design

For the abovementioned objectives, we used a nonexperimental correlational design divided into two parts: an exploratory factor analysis (EFA) for describing the factorial loads of items corresponding to the SQ measures and a correlational design between SFB and SQ.

### 2.2. Participants

The participants of this study were recruited through social networks, dissemination on the official Facebook page of the School of Engineering and Business, Guadalupe Victoria of the Autonomous University of Baja California, and CETyS University, as well as via Whatsapp by direct contact, sending the link to the measurement instrument of this research. The invitation was made openly, voluntarily, and anonymously, excluding the invitation to only those who had received a diagnosis of respiratory disease derived from COVID-19, were in mandatory isolation, and had been pregnant for at least six months. On the other hand, the sampling was performed in two stages. Two-step samples were used for this study. An initial sample of 1667 participants who met the necessary criteria (Figure 1) for exploratory and confirmatory factor analysis was obtained. To analyze the effects of SFB on SQ, a total of 535 participants who reported having children over 12 months old but under 12 years old at home were included. The questionnaires were published by social media. The participants had a mean age of 36.7 years (SD = 7.73), and all were over 18 years old. A pregnancy or labor at least six months ago or having received a diagnosis of respiratory disease derived from COVID-19 and being under mandatory isolation were exclusion criteria for participation, and participants with missing data were excluded from further analysis. The participants’ demographic characteristics are shown in Table 1.

### 2.3. Instruments

The negative effects of social isolation during the COVID-19 pandemic were measured with the SFB and SQ questionnaire [1], an online-applied instrument designed to measure changes in four dimensions of socio-familial behavior: emotional family-related changes, parents’ self-perceived behavior changes, changes in the family’s home activity habits, personal perceived effects, and personal and familial perceived effects during social isolation due to COVID-19, and its effects on SQ. This questionnaire is composed of two parts, part one comprises 22 items with four response options scored using a Likert scale from “In disagreement” (1) to “Strongly agree” (4). For more information, please consult the paper indicated as [1]. This questionnaire can be consulted as a Appendix A. For part two, measures about changes in sleep quality, an ad hoc questionnaire was developed, incorporating three theoretical factors used as reference studies of voluntary social isolation related to the effects on sleep [28]. This section comprises 13 items that were designed to explore changes in sleep habits related to schedules, the perception of sleep quality, and changes in sleep habits related to consumption of carbonated and energy drinks. In the original instrument, nine of the items were scored using a dichotomous scale with yes/no response options, while three questions were related to the number of hours of sleep or the time of going to sleep, which were openly answered on a numerical scale. Finally, one question, “As a result of voluntary isolation, how do you consider your quality of sleep?” This question was scored using an ordinal scale from “very good” to “very bad”. Responses with non-dichotomous options were transformed as follows: The answers to the question “Specify the approximate time you go to bed during the week, since the outbreak began” were assigned a value of 0 when the hours were between 8:00 p.m. to 12:00 a.m. and a value of 1 when the hours were outside that range. The answers to the questions, “Please specify the average number of hours you regularly sleep each night” and “How many hours per day (including nap times) do you currently sleep on the weekends?” were assigned a value of 0 when the range of hours of sleep was between 7 and 9 and a value of 1 when they were outside of it.

### 2.4. Procedure

The questionnaires were openly published by social media and were available for two weeks (21 April 2020 until 5 March 2020) until the completion of the study. The respondents were instructed to answer as honestly as possible and to indicate participation only if they were living under voluntary isolation due to the outbreak of COVID-19. The completion of the questionnaire took approximately 12 min. Responses were automatically sent to a native Excel database, which was later manually recoded into numerical values and categories.

### 2.5. Data Analyses

After the removal of individuals who did not meet the required criteria for the study, the instruments were processed separately: (1) an SQ EFA based on the tetrachoric correlations for describing its reliability, and the final items were selected for inclusion in the instrument and statistical analysis and (2) a linear regression model was analyzed to describe the possible predictive association of the negative effects of social isolation during the COVID-19 pandemic on SFB and sleep quality. The model was customized by using sleep quality as the dependent variable and SFB as the independent variable. The dimensions were classified, by means of procedures [1] reported elsewhere (the sum of correlated groups of items), into six main groups of total scores: the SF: Personal score, SF: Family score, SF: Parents’ score, SF: Habits score, SF: Emotional score, and SF: Total Score, while SQ remained a unique dimension according to the first step and used the enter method for each variable. All data were analyzed using Factor Analysis Program Software Package version 12.01.02 (Rovira I Virgili University, Tarragona, Spain) [33] and SPSS 25. Reliability estimations were calculated from the polychoric correlation matrix using factor analysis.

### 2.6. Ethical Considerations

The Ethics Committee of the Faculty of Engineering and Business, Guadalupe Victoria of the Autonomous University of Baja California accepted the ethical considerations for the protocol of this study (registration number: POSG/020-1-01). All procedures of the study considered the Helsinki Declaration and agreements. No questions regarding personal identification data were asked in this study. The participants did not receive any compensation for their collaboration.

## 3. Results

### 3.1. Reliability Measures of Sleep Quality

For interpretation purposes related to sleep quality, each item was dichotomized to values 1 and 0. The reliability of the SQ section of the test was carried out in two parts: the EFA and CFA. An initial subsample of 700 participants was used for the EFA using a model that was first composed of 15 items in the first iteration, which suggested a unique dimension. We performed a second analysis in which loadings > 0.40 were excluded. The output values conserved the same direction indicating the presence or absence of symptoms that interfere with sleep quality, final included items with a 5.0 eigenvalue for the reliability tests are shown in Table 2. Bartlett’s sphericity test was 2003.9 (*df* = 45, *p* < 001), the good sample size was good, and the Kaiser–Meyer–Olkin score of 0.82 was adequate. The test had a Cronbach’s alpha of 0.905 and McDonald’s omega = 0.907.

A total subsample of 713 participants was employed to carry out the CFA, in which items showing larger errors and lower factorial loadings were excluded. The final model consisted of items 1, 2, 3, 7, 9, 11, and 13 with a higher parsimonious index (Table 3).

### 3.2. Effects of Socio-Familial Behavior on Sleep Quality

The main regression results are shown in Table 4. For this analysis, one model was entered using the global score of sleep quality as the dependent variable and the socio-familial behavior SF: Personal score, SF: Family score, SF: Parents’ score, SF: Habits score, SF: Emotional score, and SF: Total Score as the independent variables (*R* = 0.497, *R*^2^ = 0.247, F = 41.76, *p* < 0.001). Two predictive factors, SFB-Habits and SFB-Emotional, showed the main effect over the SQ measures with the highest correlations. Partial regression plotting for the significant predictive factors is shown in Figure 2A,B, showing only a strong significant association for items related to habits and perceived emotional changes of the participants with changes in their sleep quality. The descriptive statistics for all variables are shown in Table 5, in which higher scores represent higher perceived changes for SFB and poorer SQ Pearson’s bivariate correlations were significant between the SFB: Habits score and SQ (*r^p^ =* −0.231, *p* < 0.001) and the SFB: Emotional score and SQ (*r^p^* = 0.423, *p* < 0.001).

## 4. Discussion

Facing pandemic social isolation burdens and threatens mental health stability. The global COVID-19 lockdown involved a series of factors that still challenge the public health field’s understanding for resolving the consequences. To date, great advances have taken place; however, there are still very few studies reporting effects on mental health. This study delivered a reliable instrument for measuring SFB and its effects on SQ and considered a three-domain exploration for understanding changes in behavior during social isolation as well as its effects on SQ. The data presented in this study support the hypothesis that changes in SFB negatively affect SQ, particularly for emotional familial factors, including the ability to self-control emotions and control family and parent–child interactions. The positive correlations observed in these results suggest that an increase in emotional changes reduces SQ.

Previous studies have reported that lockdowns during the COVID-19 pandemic were not a factor for changes in sleep quality, indicating that adolescents did not seem to substantially modify their sleep habits [34]. However, the sampling and inclusion criteria should be carefully considered for any sleep and SFB descriptions, as sleep habits are closely related to family structures [35], which may vary from the procedures implemented in the present study.

This study suggests that mental health during social isolation fur to COVID-19 is affected not only by social isolation, as reported elsewhere [36] or by intrafamily members’ habits, such as eating [37] but also by familial interactions, the resolution of household needs, or schedules for daily life activities, which is consistent with the findings in the present study. Family habits and emotional changes have been shown to have an influence on SQ, including schedules and substance use for sleeping. Other studies have described that personal familial perceptions of the pandemic produce anxiety specifically in mothers, causing insomnia and potentially changing children’s sleep quality [37]. During the pandemic, social isolation produced an increase in family member interactions once it became the only social activity for this particular population. Our study revealed that sleep quality is related to the ability to resolve daytime emotional needs and maintain better-organized habits at home.

COVID-19 has led to generalized stress and anxiety conditions among populations due to uncertainty and feeling overwhelming, and people face the direct effects of social changes in public services such as the maintenance of law and order [37]. However, families in general could be strongly susceptible to the stress- and anxiety-related effects of social isolation, even when family members are not infected; daily life modifications and the perception of an increase in activities, such as education at home, economic needs, or limitations of certain basic goods, can be related to sleep quality.

The literature suggests that sleep disorders appear when there is a gap between internal and external synchronizers of sleep, causing changes in regular sleep patterns and a series of subsequent symptoms [37]. Different diagnostic criteria have been proposed for these disorders, but in general, they consist of a persisting inability to fall asleep at earlier, more desirable and socially conventional times at night, accompanied by extreme difficulties waking in the morning, causing problems in the daily performance of the individuals who suffer from untreated sleep disorders [37]. Multiple studies have shown that circadian rhythm sleep disorders are commonly associated with depression and anxiety symptoms and represent a major risk factor for one or even both of these disorders in the future [38,39].

Further analysis is needed to explore the possible influence of other studied factors, such as the direct influence of the increase in alcohol consumption and economic burdens, as the descriptive data of this study showed a considerable proportion for both conditions.

## 5. Conclusions

The main contribution of this study was describing three predictive factors involving the negative effects of SFB measures on SQ. In addition to the global SFB score, the Habits, Emotional, and Parents’ scores, as subdivided into the main modified domains during the COVID-19 outbreak, were demonstrated to have a significant influence on SQ. Data analyzed from SFB changes during self-isolation due to the pandemic suggest significant associations between sleep quality and SFB experiences; positive correlations suggest an increase in negative effects on sleep quality accompanied by negative socio-familial experiences in different dimensions. Sleep quality, schedules, substance use, and the total sleep quality score were associated with the personal perception of abilities to face problems and hope for the future. We also identified an affected family behavior of the perception of abilities to resolve conflicts with children at home, to perceive stress and concerns about family members, and to resolve feeling overwhelmed by school. Habits in the abilities to make schedules at home with family members were associated with an effect on sleep quality for all the studied dimensions. Future studies should consider differentiating socioeconomic and economic activities, as well as the history of sleep schedules and SFB among families, as well as being sensitive to the sex of the respondents.

## Figures and Tables

**Figure 1 ijerph-19-03702-f001:**
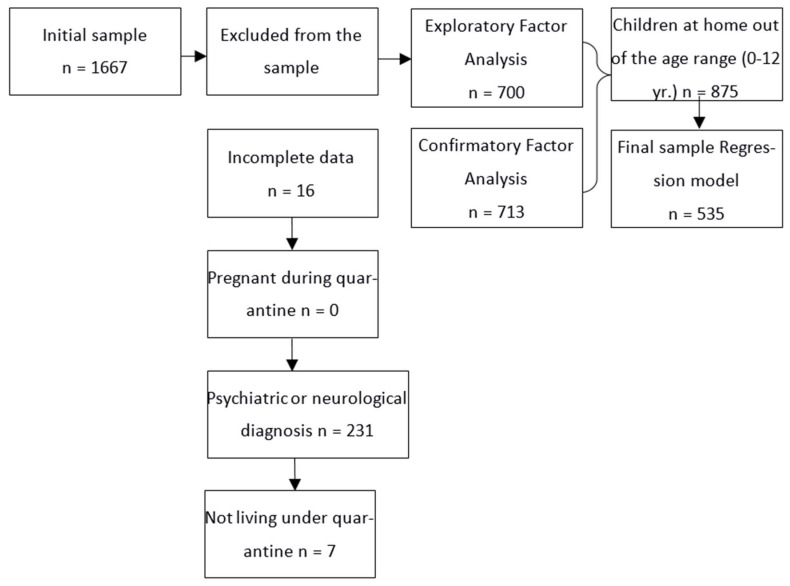
Steps for data analysis and exclusion of participants who did not meet the inclusion criteria.

**Figure 2 ijerph-19-03702-f002:**
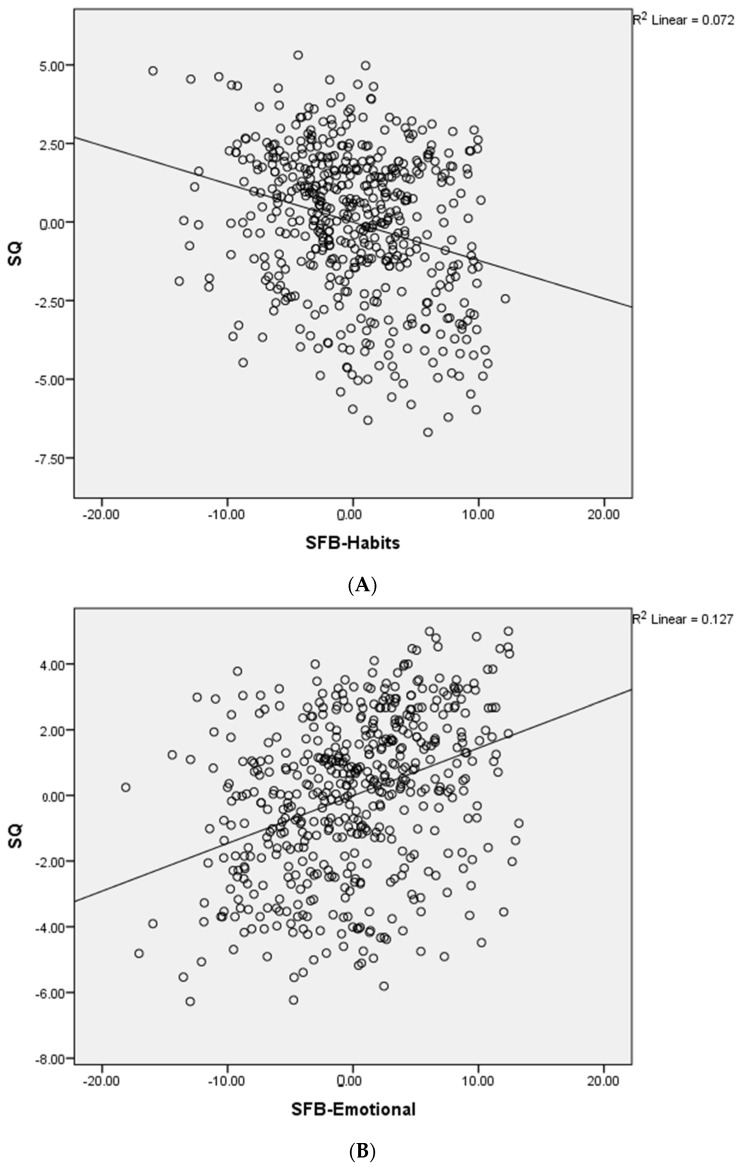
(**A**,**B**). Partial regression plots using sleep quality as the dependent variable; Note: SQ, sleep quality. SFB-Habits, socio-familial habits. SFB-Emotional, socio-familial emotional.

**Table 1 ijerph-19-03702-t001:** Demographic characteristics of the sample.

Age	Mean 36.7 Years (SD = 7.7)				
>Sex	18.5% male		81.1% female	0.4% preferred not to answer	
Marital Status	12% single,	60.7% married,	18.3% free union	8.2% divorced	0.7% widower
Health Service Benefits	50.1% public service		18.5% private service	24.9% both public and private services	
Smoking Status	14% smokers		86% non-smokers		
Increase of Alcohol Consumption	26.2% increased		73.8% not increased		
Body Weight	73.4 kg (SD = 16.3)		Range 110 kg (150 kg max/40 kg min)		
Economic Concerns as an Effect of Social Isolation Regulations in the Country	76% yes		24% no		

**Table 2 ijerph-19-03702-t002:** Factorial loadings, communalities, and confidence intervals from the tetrachoric matrix for sleep quality measures.

					95% CI
Factor	Value 1	Value 0	Factorial Loadings	Communalities	[LL, UL]
Since the beginning of the outbreak, do you believe that your sleep habits have changed?	Yes	No	0.907	0.822	[0.855, 0.956]
Please specify the average number of hours you regularly sleep each night.	<6 h, >10 h	7–9 h	0.592	0.35	[0.538, 0.674]
Since the beginning of the outbreak, have you noticed that you go to bed later than before the outbreak?	Yes	No	0.848	0.719	[0.803, 0.899]
Specify the approximate time you go to bed during the week since the outbreak began.	2–6 h later	1–2 h later	0.719	0.517	[0.682, 0.781]
As a result of voluntary isolation, how do you consider your quality of sleep?	Regular, bad, very bad	Very good, good	0.709	0.503	[0.669, 0.758]
How many hours per day (including nap times) do you currently sleep on the weekends?	<6 h, >10 h	7–9 h	0.784	0.615	[0.728, 0.873]
In the past month, have you been diagnosed with a medical illness that relates to sleep disorders (mostly mental health issues such as depression or anxiety)?	Yes	No	0.427	0.182	[0.384, 0.524]
Since the beginning of the outbreak, have you perceived that you wake up at night to go to the restroom or for no reason?	Yes	No	0.551	0.304	[0.513, 0.610]
Since the beginning of the outbreak, do you feel rested when you wake up?	No	Yes	0.501	0.251	[0.447, 0.589]

Note: 1 = corresponds to affection to SQ symptoms, 0 = represents absence of affection of SQ symptoms.

**Table 3 ijerph-19-03702-t003:** SQ Intervariance measures.

Items	Eliminated Items	χ^2^ (*df*)	*p*	χ^2^/*df*	RMSEA	SRMR	CFI	GFI	PCFI
10	non	643.60 (35)	<0.001	18.38	0.15	0.09	0.68	0.84	0.53
9	14	501.93 (27)	<0.001	18.59	0.15	0.08	0.72	0.86	0.54
8	4	180.03 (20)	<0.001	9.00	0.10	0.06	0.86	0.93	0.61
7	15	68.38 (14)	<0.001	4.88	0.07	0.04	0.94	0.97	0.62
6	13	33.9 (9)	<0.001	3.76	0.06	0.03	0.97	0.98	0.58
5	2	11.98 (5)	<0.001	2.39	0.04	0.03	0.99	0.99	0.49

**Table 4 ijerph-19-03702-t004:** Linear regression results for the effect of socio-familial behavior under social isolation during the COVID-19 pandemic dimensions on sleep quality.

Predictor	*b*	SE	*beta*	t	*p*	(95%) CI	*r*	Partial	Part	Tolerance	VIF
SF: Total Score	0.147	0.054	0.108	2.714	0.007	[0.041, 0.254]	0.187 **	00.119	0.104	0.930	1.075
SF: Parents’ score	0.027	0.035	0.033	0.754	0.451	[−0.043, 0.096]	−0.132 **	0.033	0.029	0.773	1.293
SF: Habits score	−0.131	0.022	−0.266	−5.973	0.000	[−0.174, −0.088]	−0.317 **	−0.256	−0.230	0.746	1.341
SF: Emotional score	0.144	0.017	0.347	8.538	0.000	[0.111, 0.177]	0.424 **	0.354	0.328	0.895	1.117

Note: SF, socio-familial, *b* represents unstandardized regression weights. SE indicates standard errors. *beta* indicates standardized regression weights. *r* represents the zero-order correlation. *LL* and *UL* indicate the lower and upper limits of the confidence interval, respectively. ** indicates *sig.* Pearson correlations < 0.001. VIF, variance inflation factor.

**Table 5 ijerph-19-03702-t005:** Descriptive statistics for socio-familial behavior and sleep quality measures.

Scores	Min	Max	Mean	SD
SFB: Parents	5	20	14.73	3.26
SFB: Habits	2	32	20.50	6.15
SFB: Emotional	8	36	22.44	6.31
SFB: Total	23	123	76.60	13.36
SQ: Schedule	0	9	5.21	2.23
SQ: Quality	0	6	2.45	1.60
SQ: Substance use	0	4	1.30	0.95
SQ: Total	0	18	8.64	3.61

## Data Availability

The gathered and analyzed database for this research is available at: https://www.researchgate.net/publication/359327447_DATA_Effects_of_Socio-Familial_Behavior_on_Sleep_Quality_Predictive_Risk_Factors_in_Individuals_under_Social_Isolation.

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
