# Peer review of "Effects of Socio-Familial Behavior on Sleep Quality Predictive Risk Factors in Individuals under Social Isolation"

_ijerph, 2022, doi:10.3390/ijerph19063702_

Round 1
Reviewer 1 Report
Authors have detailed for comments and now the article appears more comprehensive. Statistical analysis allows a reliable picture on the investigated effects during the pandemic.
Thank you for the opportunity to review!
Author Response
We thak the reviewer's suggestions and contributions for the improvement of this manuscript. We have been glad on working over such a qualified and academic experience.
As the general suggestion about English language and style, we have performed a second round of improvement of English style, expecting it is more active and academic now.
Reviewer 2 Report
The overall presentation seems improved. Response to my prior reviews is generally quite good. Also, the change in article title is much appreciated--this new title better reflects what is covered in the article.
I just have a couple minor editorial notes:
- Line 61 -- instead of "economic burden" wouldn't this be "economic concern" (to match the change made elsewhere in this terminology)
- Lines 135-139: Some of this is stated in the new wording just above and can be deleted here. I would delete on lines 135-136: "the questionnaires were published by social media." I would delete on lines 137-139 the wording from "A pregnancy or labor..." up through "...criteria for participation, and". I would keep the note about missing data.
Author Response
|
Reviewer comments |
Response |
|
Line 61 -- instead of "economic burden" wouldn't this be "economic concern" (to match the change made elsewhere in this terminology)
|
Thank you for the suggestion, indeed, to use this name for the variable in a coherent way among the manuscript. The term has been changed accordingly. |
|
Lines 135-139: Some of this is stated in the new wording just above and can be deleted here. I would delete on lines 135-136: "the questionnaires were published by social media." I would delete on lines 137-139 the wording from "A pregnancy or labor..." up through "...criteria for participation, and". I would keep the note about missing data. |
Overall comments and suggestions made by the reviewer helped to improve the quality of this manuscript. The suggestion has been applied as well. |
|
General suggestion about the language and style. |
We performed a second round of improvement of English style, expecting it is more active and academic now. |

This manuscript is a resubmission of an earlier submission. The following is a list of the peer review reports and author responses from that submission.
Round 1
Reviewer 1 Report
In this study, the authors studied the risk factors for sleep quality due to changes in socio-family behavior during social confinement by the COVID-19 outbreak.
I have a few concerns,
- Please provide a table in the main text to show the summary of demographic characteristics in the groups.
- Please provide a flowchart to show how the participants were included and what are the filtering steps to remove individuals that are not meet the requirement.
- The authors may provide the questionnaires as a supplementary file.
- In section 2.5, it is too general for the description of data analysis, we need to know more details, what are the models, want are the inputs, how is the analysis formula designed. Please provide as detailed as you can, this is the most important part of this part, otherwise, this manuscript cannot be published with detailed methods missing.
Author Response
We appreciate the accurate revision of our manuscript as well as all suggestions to improve it. In the following section a brief description of the changes that were applied to the manuscript is presented.
Comment 1. Please provide a table in the main text to show the summary of demographic characteristics in the groups
Decision made: A table with demographic characteristics was included in the participants section.
Comment 2. Please provide a flowchart to show how the participants were included and what are the filtering steps to remove individuals that are not meet the requirement.
Decision made: 2.2 A flowchart for describing removal of participants who did not met the inclusion criteria was included.
Comment 3. The authors may provide the questionnaires as a supplementary file.
Decision made: Questionnaires are now submitted as recommended.
Comment 4. In section 2.5, it is too general for the description of data analysis, we need to know more details, what are the models, want are the inputs, how is the analysis formula designed. Please provide as detailed as you can, this is the most important part of this part, otherwise, this manuscript cannot be published with detailed methods missing.
Decision made: Details about the statistical analyses were added.

Reviewer 2 Report
The article presents a hot topic arguing with mathematical tools certain conclusions perceived at broad level by individuals.
Details about the questionnaire may improve the understanding. In fact, no information about the questionnaire structure and the responses distribution are given.
The results should be commented as the figures without comments appear less relevant. The discussion section should be connected with the reported results. The article connects with other references, some comments are general perceptions, but these should be connected with the implemented study.
Author Response
We kindly appreciate comments and suggestions to the present study. The decision made for each one of them is attached in the submitted file.

Reviewer 3 Report
This manuscript reports results of a convenience sample survey regarding impacts of (COVID 19) isolation on social behavior and sleep quality.
The manuscript is not very well written and is very difficult to follow. Very limited information is provided on the methods and results (making the discussion and conclusions difficult to follow).
In the abstract, the authors should focus more clearly on the overall results and less on the technical presentation of methods/analysis. As it is, the methods/analysis are not clearly presented throughout the manuscript.
- "Observational transversal research design" is not an overly common term--this should be explained in clearer language.
- More information on how the sample was solicited is needed. What population was targeted? What geographic area?
- More information is needed on the specific measures used. How is the sleep quality survey ad hoc? What are the items/wording that are reported in the factor analysis?
- There is very little information presented that can help the reader understand the methods and resulting analysis/results.
Author Response
We kindly appreciate reviewer's comments and suggestions. In the attached file, the decision made to each one of them is described.

Round 2
Reviewer 1 Report
The authors have addressed my concerns. One more comment: Please refine Figure 1 and Table 1, they are of low quality in the current format.
Reviewer 2 Report
The article has been improved by adding details, results and comment.
It may be published as it is.
Reviewer 3 Report
The manuscript still needs considerable work in terms of writing quality--it remains difficult to follow. There are still numerous issues with the presentation of information.
While the addition of measurement information is helpful, the authors should still explain how they went about soliciting their sample a bit more. What social media platforms were used; how were people asked to participate and so on. It is still too vague to understand who this might include.
In Table 1, it is unclear what the variable "economic burden" means.
In the supplementary file, there is a single questionnaire but the authors keep referencing two questionnaires in the text. This needs to be clarified.
The sample size is 535, so how did they get two subsets that are each at least 700 to do the EFA and CFA? (referred to on lines 220 and 238)
The factor analysis reported in Table 2 indicates the use of items that do not match in terms of their response categories. This is somewhat problematic, but what may be even more of an issue is that the last item was likely missing data. If they answered no to the previous question, were they not included in the analysis? How could a factor analysis be run with incomplete information?